# Interfacial Adhesion in Silica-Silane Filled NR Composites: A Short Review

**DOI:** 10.3390/polym14132705

**Published:** 2022-07-01

**Authors:** Chang Seok Ryu, Kwang-Jea Kim

**Affiliations:** 1School of Chemical Engineering, Chonnam National University, Gwangju 61186, Korea; netryu@hanmail.net; 2DTR VMS Italy S.r.l., 25050 Brescia, Italy

**Keywords:** silica, silane, natural rubber (NR), hydrolysis, interfacial adhesion, zinc mechanism

## Abstract

We reviewed the accelerators, the hydrolysis and condensation reaction mechanism of bifunctional alkoxy silane, and the mechanism of zinc ion in natural rubber (NR) composites. NR composites transform into thermoset composites after vulcanization reaction with help of sulfur and accelerators. Bifunctional alkoxy silanes chemically bond between NR and inorganic silica. For alkoxy silane coupling with silica surface, hydrolysis reaction takes first and then condensation reaction with hydroxyl group in silica takes place. With help of zinc ion the reaction efficiency increases significantly. Zinc ion, a smart material that increases accelerator synergy, mechanism for improvements of interfacial adhesion between NR and silica was revisited.

## 1. Introduction

Silica-silane filled natural rubber (NR) composites, as a tire tread material, are currently used in various fields as a smart composite replacing the conventional carbon black (CB) tire composites. Compared with CB filled NR tire, it shows excellent performance such as improved fuel efficiency due to low rolling resistance, reduced braking distance due to slip resistance due to high tan delta, and improved traction in snow and wet conditions when driving a vehicle. Currently many vehicles use a green tire (silica filled tire) [1]. It is called green tire because it has green strength during silica composite processing. The green tire composite has differences in interfacial bonding compared to the CB compound. And even now, various efforts are being made to increase the efficiency of interfacial bonding. Compared with the CB filled NR composite system, in both systems the interfacial interaction in the silica-silane filled NR composite has the same mechanism for chemical bonding of rubber by sulfur, but the interfacial interaction between filler and NR is different. That is, the CB filled system forms a bond by interaction between CB and NR by increasing compatibility due to the π-π interaction between the double bonds of the two materials. However, in the silica filled system, as bifunctional silane chemically bonds silica and NR, the interfacial interaction between the two materials increases by chemical bonding.

Until now, there was no published article introducing the mechanism for the entire process of the chemical reaction from the raw material to the silica-silane-NR composite to increase the interfacial interaction. This paper briefly summarizes the whole process above. We first reviewed the accelerators that affect the degree and rate of bonding between rubber chains, and reviewed the hydrolysis mechanism of alkoxy silane and the condensation reaction mechanism between hydrolyzed silane and hydroxyl group on silica surface. And the chemical reaction mechanism between bifunctional silane and rubber was reviewed. Zinc ion, a smart material that effectively increases accelerator synergy and increases the effectiveness of vulcanization, was also reviewed.

## 2. Materials and Methods

### 2.1. Accelerators

Various additives are added to rubber composite such as filler, plasticizer, compatibilizer, anti-degradant, processing aid, curative, accelerator, etc. A typical curative used in rubber is sulfur, which chemically promotes bonding between rubber chains to increase interfacial adhesion between chains. Accelerators play a role in controlling the degree and rate of chemical reaction between sulfur and rubber chains. Cure time or degree of crosslinking is different depending on the type of accelerator thus the choice of accelerator is different depending on the composite and processing purpose. First, we revisit the history of accelerators.

Goodyear-Hancock first showed unaccelerated vulcanization of rubber [2,3]. Later, organic accelerator, aniline, was first introduced in 1906 [4]. However, aniline was too toxic for use in rubber vulcanization. Later, reaction product of aniline with carbon disulfide such as thiocarbanilide was introduced in 1907 [5]. Subsequently carbon disulfide modification with aliphatic amines (e.g., dithiocarbamates), which thiuram type accelerators were introduced in 1919 [6]. And then delayedaction accelerators such as 2-mercaptobenzothiazole (MBT) and 2-mercaptobenzothiazole disulfide (or 2,2′-dithiobisbenzothiazole) (MBTS) were introduced in 1925 [7,8,9]. The first commercial benzothiazole sulfenamide (N-cyclohexylbenzothiazole 2-sulfenamide), a delayed-action accelerator, was introduced by Harman in 1937 [10]. More delayed-action premature vulcanization inhibitor (PVI), N-(cyclohexylthio)phthalimide (CTP), was introduced in 1968 [11]. Studies on accelerator effects on vulcanization were focused mainly on CB filled systems. Various types of accelerators and their roles in CB filled rubber compounds were summarized [6]. Thurn et al. [12] first discovered the use of silica in combination with bis(3-triethoxysilylpropyl) tetrasulfane (TESPT) and NR, and this was patented for practical ‘green tire’ application in 1992 [13]. Since then, considerable efforts have been made on studies of silica-silane reinforcement in rubber compounds [14], silica dispersion [15,16,17,18,19], silica-silane reaction [15,16,20,21,22,23,24,25,26,27,28], silane-rubber reaction [29,30,31], zinc ion- [26,32,33,34,35], moisture- [36,37], temperature- [38], and accelerator effects on vulcanization properties [39,40], effects of processing geometry [19], and effects of polymer blends [27,28,41,42].

There were many studies on accelerator effects on vulcanization properties of CB filled rubber compounds [39,40,43,44,45]. For example, thiazole type accelerators on crosslinking rate in the NR compound [43] and NR & synthetic poly-1,5-dienes [44] were studied. Sulfenamide type accelerators on rate of sulfenamide-sulfur reaction [45] and scorch delay reaction in the NR & SBR compound [46] were studied. Vulcanization rate of silica/NR compounds and CB/NR compounds, which contain various accelerators were directly compared [1,35,40]. These accelerators consist of different chemical structure, i.e., thiuram type, TMTD (tetramethylthiuram disulfide) and DPTT (dipentamethylenethiuram tetrasulfide); thiazole type, MBT (2-mercaptobenzothiazole) and MBTS (2,2dithiobis(benzothiazole)); sulfenamide type, CBS (n-cyclohexylbenzothiazyl-2-sulfenamide) and NOBS (n-oxydiethylenebenzothiazyl-2-sulfenamide)]; and zinc ion containing thiuram type ZDBC (zinc di-n-butyldithiocarbamate)]. The order of accelerator reaction was thiuram(TMTD,DPTT), thiazole(MBT,MBTS), and then sulfonamide(CBS,NOBS) types due to their chemical structure [1].

### 2.2. Silica vs. CB

Comparing silica surface with CB, silica surface consists of Si-O bonds with hydroxyl groups (-OH), while CB mainly consists of C-C and C=C bonds. Silica particles exhibit a strong filler-filler interaction due to its polar (Si^+δ^-O^−δ^) character due to differences in electro negativity between Si(3.5) and O(1.9) thus easily agglomerate each other, while CB shows a weak C=C interaction due to the presence of a π-π bond, thereby showing good compatibility with rubber chains, which contain a double bond. The π-π bonds and/or oxygen on the CB surface interact with double bonds (π-π) in the rubber chain, which shows good compatibility with the rubber chain. Therefore, the dispersion of CB is better than that of silica in the rubber matrix. On the other hand, modification of the silica surface with silanes, particularly with bifunctional organosilanes, such as bis(triethoxysilyilpropyl)disulfide (TESPT) and bis(triethoxysilyilpropyl)tetrasulfide (TESPD), assists in the dispersion of silica agglomerates in the rubber chain [18,24]. The chemical reaction between silica and silane as well as silane and rubber modifies the network of the rubber matrix [23]. It has been known that hydroxyl group on silica surface chemically reacts with hydroxyl group in hydrolyzed silane via condensation reaction, and then forms a 3-dimensional (3D) network structure with a rubber chain [15,16,23]. Hence, the filler network constant (filler network interaction parameter) of silane-modified silica is different from CB filled systems. This makes an interpretation of the in-rubber structure of the modified silica system more difficult.

In silica/NR composites, the order of accelerator reaction was the same as shown in the CB filled system, i.e., thiuram (TMTD, DPTT), thiazole (MBT, MBTS), and then sulfonamide (CBS, NOBS) types due to their chemical structure. However, silica/NR composites showed a slower vulcanization rate (ts12, t10, t90) than the CB/NR ones for each accelerator. This is due to two stage reaction in silica-silane/NR composite, which are hydrolysis of alkoxy silane and condensation reaction between hydrolyzed silane and silica surface [1].

### 2.3. Interfacial Interaction

For, interfacial interaction, Wolff and his coworkers introduced the in-rubber filler structure (αF) for quantitative measurements of the filler structure [47,48,49,50,51]. Wolff et al. [50] reported that the αF was constant over the entire range of carbon black filled in NR and NBR systems. Their research focused mainly on the filler-filler interaction in rubber compounds. On the other hand, in the case of a silica filled system, αF increased with increasing filler concentration. Kim reviewed his research on the in-rubber structure of CB and silane-treated silica [52].

### 2.4. Hydrolysis Mechanism

The hydrolysis mechanisms as well as the hydrolysis measurement technique and its practical applications in manufacturing fields are revised by Kim [53,54]. This section reviews the theoretical aspects of acid-catalyzed and base-catalyzed hydrolysis mechanisms.

#### 2.4.1. Acid Catalyzed Hydrolysis

Hydrolysis means that moisture is applied to molecular chains, causing the chains of molecules affected by moisture to break down. Moisture present in the composite affects hydrolysis. When in contact with high-purity moisture in a neutral, low ionic state in a non-glass container, silane maintains a stable state for several weeks or months. However, when in contact with tap water, the hydrolysis of alkoxysilane proceeds considerably within a few hours [55]. The catalyst applied to the hydrolysis of alkoxysilane is also applied to the progress of the condensation reaction, and the waterborne silane improves adhesion to the glass surface than the control silane [55]. This means that the catalyst present in moisture significantly increases the silane bond on the silica surface. The primary reaction increases the silanization reaction in the presence of moisture, and the secondary reaction increases with higher moisture content [56,57,58]. Moisture on the silica surface favors the reaction between the silica and the coupling agent (e.g., TESPD, TESPT), this also reducing the interaction between silica particles [57]. The hydrolysis reaction mechanism also applies for plastics (e.g., polyamide) [59]. Figure 1a,b shows the hydrolysis mechanism by catalysis of base- and acid- of alkoxysilane. The reaction coordinate has a structure of polar parameter values (ρ*) and steric parameter values (s) from the reactant to the transition state. As the reaction proceeds, it qualitatively shows changes and charge distribution [60]. Figure 1a shows the mechanism of acid-catalyzed hydrolysis. Five substituents exist around the silicon atom in the form of SN2-Si. It forms a pentacoordinate intermediate state. The approaching nucleophile and the outgoing leaving group hybridize with the silicon atom, resulting in sp3d hybridization between the nucleophile and the remaining group. The meaning of small values of ρ* and s is shown in Figure 1a. The reaction mechanism is that after rapid equilibrium protonation in the equilibrium state of the substrate, the water molecule becomes a leaving group and a bimolecular molecule, resulting in SN2-type displacement [61,62].

#### 2.4.2. Base Catalyzed Hydrolysis

The base-catalyzed hydrolysis mechanism is illustrated in Figure 1(b). Bi-molecular nucleophilic substitution reaction (SN2**-Si or SN2*-Si) with the coordinate intermediate; SN2**-Si refers to the rate determination of the pentacoordinate intermediate formation (k1 < k2), and SN2*-Si refers to the rate determination of the break-down (k1 > k2) [61]. A large value of ρ* means that the transition state (T.S.) is more stable than the reactants by electron withdrawing groups attached to silicon atoms. In the proposed mechanism, the negative charge at the silicon atom in the transition state (T.S.1 or T.S.2) is quite high. A large steric parameter (s) means that the transition state is sterically more crowded than the reactant [62,63]. The sp3 hybridized silicon forms bonds with the introduced hydroxide anions during rehybridization, forming the same transition state as sp3d. It forms a strongly coupled transition state between the negative charge density and the nucleophilic oxygen on the silicon surface [64]. Figure 2 shows the hydrolysis mechanism according to the pH of alkoxysilane [65]. As stated by Allen model [66] and Kay & Assink model [67], bas-catalyzed hydrolysis reaction is a condensation reaction when an alkoxy group (-(OR)3) is replaced with a -(OSi)3 group (bottom left); When an alkoxy group (-(OR)3) is substituted with a hydroxy group (-(OH)3) (bottom right), this is called hydrolysis. Acid hydrolysis is significantly greater than base hydrolysis and is minimally affected by other carbon-bonded substituents. The results of Osterholtz & Pohl [68], illustrated by Moore [69], show that when the pH is neutral, hydrolysis occurs last, and it changes 10 times for every 1 pH change. The condensation reaction is also affected by pH, and the minimum reaction pH is 4. The hydrolysis reaction of alcohol is reversible and stabilizes the solution for a long time.

### 2.5. Alkoxysilane Silanization Reaction Mechanism

Bifunctional alkoxysilanes are either direct or go through two reaction steps. Figure 2 shows the first step. After hydrolysis of the alkoxysilane by water molecules, the hydrolyzed silane causes a condensation reaction with the hydroxy group on the silica surface and chemically bonds to the silica surface [70]. In the second step, the sulfur at the other end of the silane chemically reacts with the double bond of the rubber chain. In this step, the sulfur (S8) ring is opened and vulcanized together with the double bond of the rubber chain. Then, a three-dimensional network is formed between silica and rubber [22,37,38,71]. This shows that the higher the degree of hydrolysis, the higher the % of the condensation reaction.

The compatibilizer controls the phase structure by increasing the interfacial adhesion property of the two materials by minimizing the difference in the value of the Flory-Huggins interaction parameter of the two materials when mixing different polymers, and thus, the applied material is reliable. In order to increase the efficiency of interfacial bonding between the resin and the filler, an interfacial adhesive (coupling agent) is treated on the surface of the filler to enhance the interfacial adhesion with the resin to induce chemical bonding at the interface. The roles of the compatibilizer are reinforces the strengths of the two materials (polymer-polymer, or polymer-filler), compensates for the weaknesses, and serves to uniformly disperse the materials [72]. Compatibilizers are mostly classified into non-reactive compatibilizers and reactive compatibilizers. Bifunctional silane is a reactive compatibilizer, which reacts with the double bond between the surface of silica and the rubber chain to chemically bond. Table 1 summarizes the structure, vulcanization form, and application fields of silanes [73].

Figure 3 shows the condensation reaction intermediate between the hydroxyl on the silica surface of a filler (e.g., silica, soft clay) and the hydroxyl of an alkoxy group substituted after hydrolysis of a bifunctional silane (e.g., TESPT) [24,30]. Figure 3 shows that after the condensation reaction, moisture is produced as a by-product. In the production site, the generation of alcohol and moisture during green composite production is regarded as indirect evidence that the material is chemically reacting properly.

## 3. Hydrolysis Quantitative Analysis

Figure 4 shows the amount of residual alcohol that did not react in TESPD during hydrolysis stage in the composite material measured by the headspace/gas chromatographic(GC) technique [70]. The residual alcohol amount of the composite with water added to silica (open circle) before mixing was lower than that of the composite without water added (closed circle), and as the mixing time increased, the alcohol content of the residual alcohol of both composites decreased. This means that the hydrolysis reaction of silane and silica increases as the presence of moisture increases and the mixing time increases.

Figure 5 shows that the sulfur in the bifunctional silane (e.g., TESPD or TESPT) chain is separated from the bond between the sulfur chains, and the sulfur group at the end forms a covalent bond with the double bond (e.g., NR, SBR) of the rubber chain [42]. Sulfur is further activated with the help of zinc as a reaction activator [74]. That is, when silane is not added, the size of the interaction between filler-polymer(silica-rubber) (αFP) is lower than that of CB and rubber chains. However, when silane is added, a covalent bond is created between the silica surface and the rubber chain through the silane as a medium, and the bonding strength is high, so the size of the interaction (αFP: filler-polymer interaction parameter in the material) is larger than that of the CB-rubber chain.

## 4. Effect of Sulfur Concentration of Hydrolyzed Silane

Figure 6 schematically shows the types of bonding between silica and natural rubber (NR) after vulcanization of various types of silanes (e.g., triethoxysilyilpropane (TESP), bis(triethoxysilyilpropyl)disulfide(TESPT), bis(triethoxysilyilpropyl)tetrasulfide (TESPD)) [23]. After silica and silane are combined by a hydrolysis mechanism, ethanol is produced as a by-product and evaporates. Later, the sulfur attacks the double bond present in the NR chain to form a covalent bond with the rubber chain. After vulcanization, it forms a 3D structure with a different structure from silica and NR depending on the type of silane used. Since TESP has no element to react with the double bond of rubber in the molecule, it exists in a state sandwiched between rubber molecules as shown in Figure 6a. In TESPD, sulfur atom attacks the double bond in the NR chain to form a 3D structure as shown in Figure 6b. TESPT also forms a 3D structure as shown in Figure 6c with NR with the same mechanism as TESPD. Since TESPT has an average of 2 more sulfur per molecule compared to TESPD, it has a higher cross-link structure than TESPD after vulcanization.

Comparing the physical properties of the silica/NR composite treated with TESPT (S4) and treated with TESPD (S2), S4 has a higher tensile modulus and lower maximum tensile stress, and has low tear resistance than S2. At room temperature or 100 °C before and after vulcanization, S4 shows a low tan δ (delta) (=E”/E’) value due to the low E” (loss modulus). In addition, it can be observed that the tan δ value of both composite decreases as the temperature increases. When the blow out (BO) test was performed on the two vulcanized composites, the S4 composite showed lower recovery from deformation compared to the S2 one. In the HBU (heat build-up) test, S4 composite showed lower heat generation than S2. In the abrasion loss test, S4 composite showed higher loss than S2, which means that S4 has a more developed 3D network structure. However, in some of the rigid 3D structures, the S4 structure may be broken earlier than the S2 when the over critical strain is surpassed.

The results of the BO test of the silica composite treated with silane and without treatment show a lot of difference [75].

Figure 7a shows the photos of the specimen before and after the ‘Firestone Flexometer’ BO test. Figure 7b shows the deformation ratio (%) of the picture above (Figure 7a). After the BO test compared to the test specimen (No. 0), the NR specimens with only silica (1&2) shows about 6-8% deformation, but the specimens with silica-silane added (3&4 (TESPT(S4)), 5&6 (TESPD(S2))) shows a strain of about 1-3% as shown in the figure (Figure 7b). In addition, as the concentration of ZB increased, it shows that the decrease in strain and increase in BO time significantly. Figure 7c shows that the addition of zinc ion to the silane/NR composite affects the increase in BO time. It also shows that the addition of ZB played a role in maximizing the above two properties.

Figure 8 shows the SEM image of the direct 3D network structure between silica-silane and NR [52]. Figure 8a shows a photograph of SEM observation that the silica surface does not have interfacial interaction with NR; however, when silane (TESPT) is added, interfacial adhesion increases between the silica surface and NR($\alpha_{FP}^{}$) as shown in Figure 8b.

The interaction parameter in the composite material is represented by $\alpha_{C}^{}$, which is expressed as $\alpha_{C}^{}=\alpha_{F}^{}+\alpha_{P}^{}+\alpha_{FP}^{}$. $\alpha_{F}^{}$ is the filler-filler parameter in the material, $\alpha_{P}^{}$ is the polymer-polymer parameter in the material, and $\alpha_{FP}^{}$ is the filler-polymer parameter in the material [52].

## 5. Zinc Ion Mechanism

Zinc ions are known to affect the improvement of the 3D network structure of the silica-silane containing rubber compound, and this is explained by the reaction mechanism of iron ions [32,76,77].

Zinc is a transition element and forms a compound with the active element in the accelerator [76]. That is, due to the ‘*d*’ orbital of the zinc ion, it forms a stable divalent complex with sulfur, nitrogen, and oxygen donor elements [76]. Figure 9a shows that zinc ions form a vulcanized pre intermediate state with donor elements such as oxygen and nitrogen [32].

The interactions of accelerators and sulfur and the crosslinking are complicated mixture of reaction sequences with kinetic and thermodynamic controlling steps [74]. Both mechanisms and kinetics seem to be controlled by the zinc compound structure. At the vulcanization stage, the zinc ion increases the NR matrix network stronger. Zinc, as a transient element, is known to form complexes with accelerators [77]. Many stable divalent complexes, with sulfur, nitrogen, and oxygen donors, can form because of full ‘d’ orbitals on zinc [76]. The complexes are typically 4, 5, or 6 coordinated. There is also evidence that π-π allyl complexes of zinc form between zinc and simple olefins [78]. The ligand complexes, a series of 4, 5, and 6 coordinated zinc structure explains the vulcanization process of all the coordinate sites on zinc and the electron nature of the carboxylate using expansion and contraction of the zinc orbitals [32]. The cross-linking mechanism with rubber matrix was explained by sulfur ring opening mechanism via bi-pyramidal model [32]. Two examples shown are for the sulfur ring opening reaction as shown in Figure 9a and the formation of the cross-linking as shown in Figure 9b. The sulfur ring opening reaction is shown as a bi-pyramidal complex with sulfur and oxygen anions, and sulfur and two amino ligands (residue from accelerators). The crosslinking intermediate allows for the transfer of sulfur from the polysulfide accelerator to the rubber in a combined mechanism. The π allylic model can be used to explain the carbon sulfur formation and types that form during vulcanization [74]. It also can be used to explain the differences in activation energies and reaction rates between various rubbers. This reaction binds some of the ZB onto the rubber backbone and creates a secondary ionic network.

Overall, “the zinc ion in zinc oxide forms a pre-intermediate state between the nitrogen (N) present in the main chain of accelerator and the oxygen (O) present on the surface of the silica to form a pre-intermediate state between the two materials. An increase in adhesion is achieved”. The evidence for this is the strong interfacial adhesion between NR and silica as shown in the SEM picture in Figure 8 and the result of the increase in physical properties as shown in Figure 7b,c.

## 6. Conclusions

The vulcanization process of silica filled NR composite is as follows. The alkoxy group at one end of the silane (e.g., TESPT, TESPD) having an alkoxy function group is hydrolyzed, and the hydrolyzed silane is subjected to condensation reaction with the hydroxyl group on the silica surface. On the other side, a sulfur function group chemically bonds with the double bond of NR to form a 3D network structure between silica and rubber. At this time, under the influence of the accelerator, the degree of reaction and the rate of reaction during crosslinking are controlled. Also, the 3D network structure is greatly increased with the help of zinc ions. Figure 10 summarizes the mechanisms involved in the bonding of bifunctional alkoxy silanes with silica and NR chains and the vulcanization of zinc ions.

## Figures and Tables

**Figure 1 polymers-14-02705-f001:**
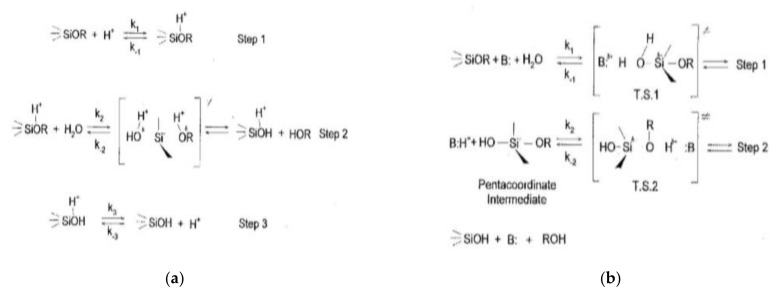
Hydrolysis mechanism for (**a**) acid catalyzed (adopted from [61,62]), (**b**) base catalyzed (adopted from [61]) systems.

**Figure 2 polymers-14-02705-f002:**
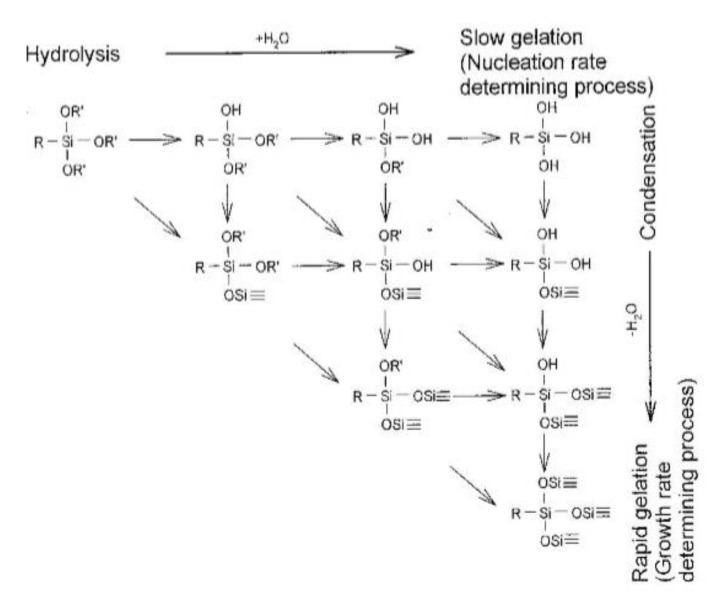
Schematic presentation of hydrolysis and condensation reaction of a trialkoxy silane; redraw from [70].

**Figure 3 polymers-14-02705-f003:**
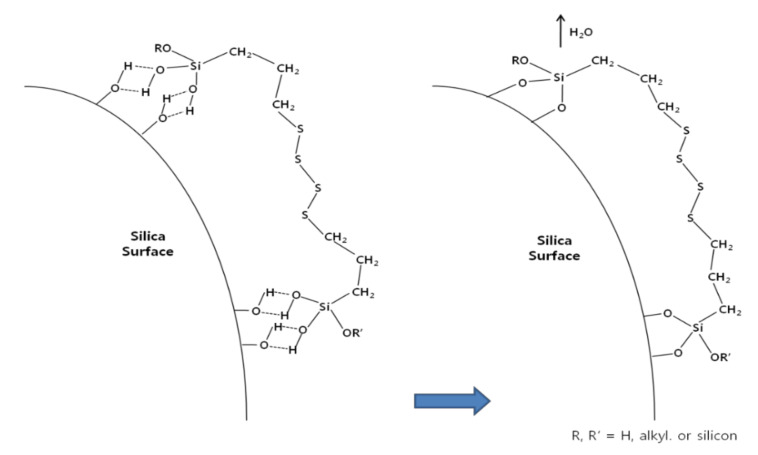
Condensation reaction intermediate of silica and hydrolyzed TESPT; redraw from [24,30].

**Figure 4 polymers-14-02705-f004:**
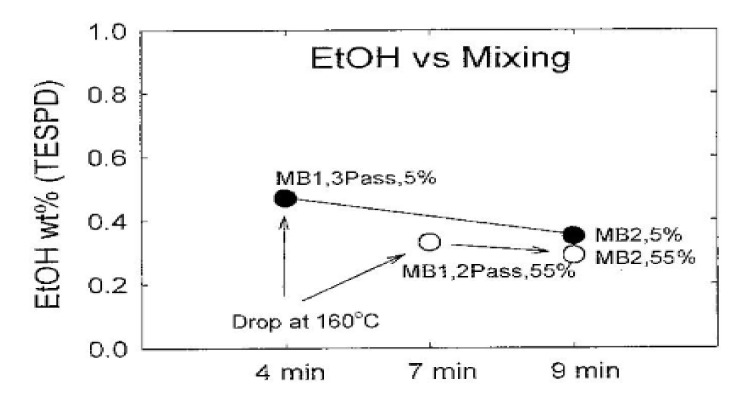
Percent of ethoxy remaining in the 3-pass vs. 2-pass compounds; adopted from [70].

**Figure 5 polymers-14-02705-f005:**
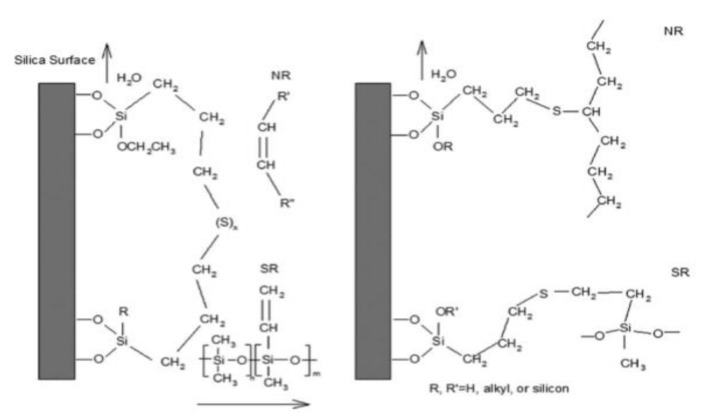
Cross-linking of unsaturated elastomers with sulfur containing silane (*x* = 2 or 4); adopted from [42].

**Figure 6 polymers-14-02705-f006:**
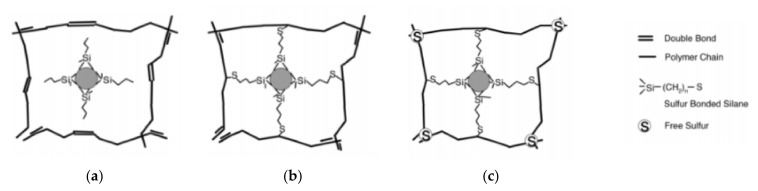
Schematic illustration of vulcanized silane treated silica (**a**) TESP (no sulfur), (**b**) TESPD, and (**c**) TESPT; adopted from [23].

**Figure 7 polymers-14-02705-f007:**
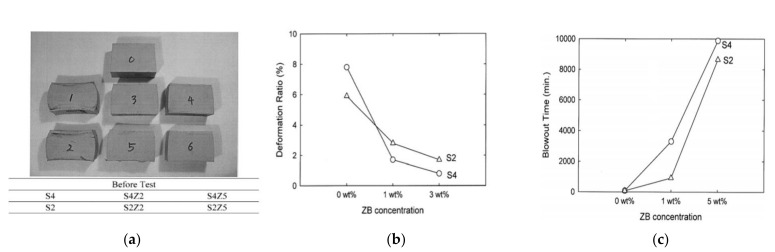
S4 (TESPT) and S2 (TESPD) compounds before and after and with/without ZB: (**a**) Photograph (0(Before Test), 1 (S4), 2 (S2), 3 (S4Z2), 4 (S4Z5), 5 (S2Z2), and 6 (SZ5)); (**b**) deformation ratio (d/D)%; and (**c**) BO time (adopted from [75]).

**Figure 8 polymers-14-02705-f008:**
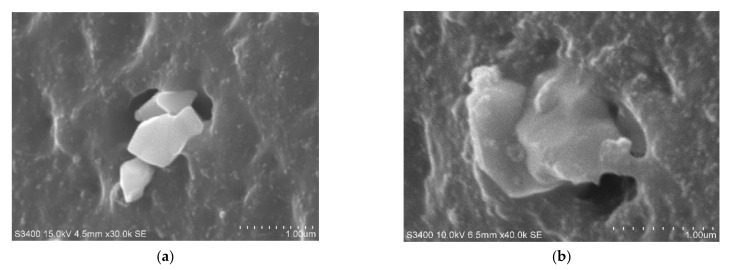
SEM photograph of silica-NR compound (**a**) without silane, (**b**) with silane(TESPT) [52].

**Figure 9 polymers-14-02705-f009:**
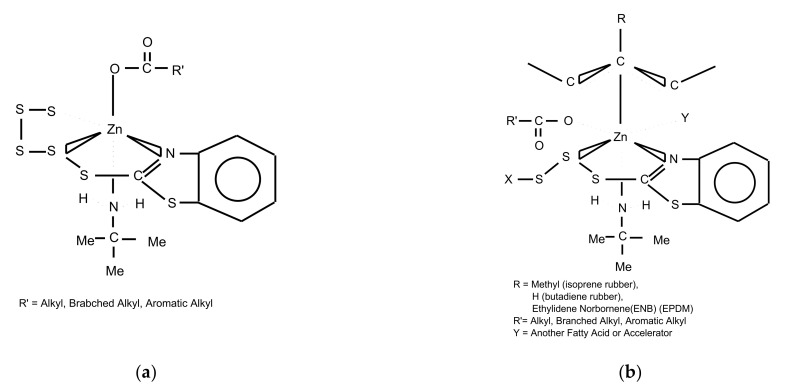
(**a**). Zinc Ion Effect on (**a**) Sulfur Ring Open Reaction Mechanism at Pre-Vulcanization Intermediate State, and (**b**) formation of the cross-linking intermediate (π allyl complex); adopted from [32].

**Figure 10 polymers-14-02705-f010:**
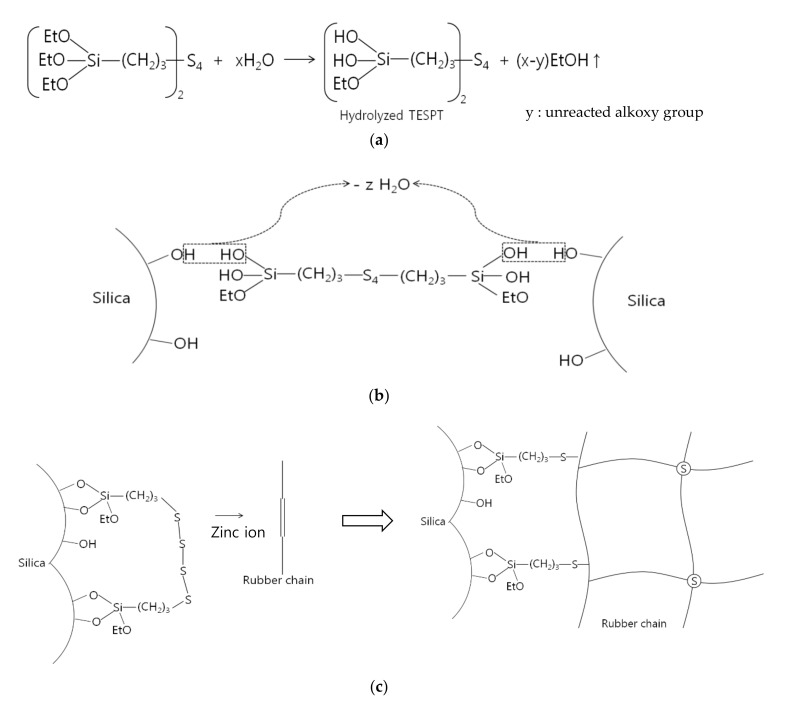
Schematic diagram of (**a**) hydrolysis of silane, (**b**) condensation between hydrolyzed silane and silica surface, (**c**) chemical reaction between sulfur and rubber chain [52].

**Table 1 polymers-14-02705-t001:** Types of silane structure, curing system and application.

Silane	Structure	Curing System	Application
TESPT	(C_2_H_5_O)_3_Si−(CH_2_)_3_−S_4_−(CH_2_)_3_−Si(OC_2_H_5_)	Sulfur	Tire treads, shoe soles, industrial rubber goods
TESPD	(C_2_H_5_O)_3_Si−(CH_2_)_3_−S_2_−(CH_2_)_3_−Si(OC_2_H_5_)	Sulfur	Tire treads, industrial rubber goods
TCPTEO	(C_2_H_5_O)_3_Si−(CH_2_)_3_−SCN	Sulfur	Shoe soles, industrial rubber goods
MTMO	(CH_3_O)_3_Si−(CH_2_)_3_−SH	Sulfur	Shoe soles, industrial rubber goods
VTEO	(C_2_H_5_O)_3_Si−CH=CH_2_	Peroxide	Industrial rubber goods
VTEO	(CH_3_−O−C_2_H_4_O)Si−CH=CH_2_	Peroxide	Industrial rubber goods
CPTEO	(CH_3_O)_3_Si−(CH_2_)_3_−Cl	Metal oxide	Chloroprene rubber
MEMO	(CH_3_O)_3_Si−(CH_2_)_3_−O−C(O)C(CH_3_)=CH_2_	Peroxide	Textile adhesion
AMEO	(C_2_H_5_O)_3_Si−(CH_2_)_3_−NH_2_	Sulfur	Special polymers, metal adhesion
OCTEO	(CH_3_−CH_2_−O)_3_Si−(CH_2_)_7_−CH_3_	Sulfur, peroxide	Processing aid

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
