# Peer review of "Interfacial Adhesion in Silica-Silane Filled NR Composites: A Short Review"

_polymers, 2022, doi:10.3390/polym14132705_

Round 1

Reviewer 1 Report

Dear Authors,

The manuscript “Interfacial adhesion in silica-silane filled NR composites: A short review” is very good. The title is OK. The abstract is short and clear. The methods are well described. Why acid catalyzed hydrolysis was used? Figure 7 a is presented in higher resolution. Explain the labels from the picture. Describe Figure 8. Explain the zinc Ion mechanism.

Author Response

Dear Reviewer 1                                                                            

Why acid catalyzed hydrolysis was used? Figure 7 a is presented in higher resolution. Explain the labels from the picture. Describe Figure 8. Explain the zinc Ion mechanism.

Following are the answers for your questions.

1. The reason why acid catalyzed hydrolysis was used is the reaction rate is faster than that of the base catalyzed hydrolysis, which industry is prefer.

2. The labels in Figure 7 are added.

Figure 7. S4(TESPT) and S2(TESPD) compounds before and after and with/without ZB (a) Photograph [0(Before Test), 1(S4), 2(S2), 3(S4Z2), 4(S4Z5), 5(S2Z2) and 6(SZ5)], (b) deformation ratio (d/D)% and (c) BO time [adopted from ref. 75].

3. Figure 8 description is added as follow;

Figure 8 shows the SEM image of the direct 3D network structure between silica-silane and NR [52]Figure 8(a) shows a photograph of SEM observation that the silica surface does not have interfacial interaction with NR; however, when silane (TESPT) is added, interfacial adhesion increases between the silica surface and NR() as shown in Figure 8(b).

4. Zinc ion mechanism is explained in mechanism section.

Sincerely

Kwang-Jea Kim Ph.D.

Reviewer 2 Report

The paper titled "Interfacial Adhesion in Silica-Silane Filled NR Composites: A Short Review", by Chang Seok Ryu et al. is, in my opinion, not sufficiently sound for publication in the journal. The following suggestions should be considered:

  1. “the filler network constant of silane-modified silica is different from carbon black filled systems.” What is the filler network constant?
  2. The sections of Interfacial Interaction is too concise, and the authors should give a brief introduction to these mechanisms.
  3. The sections of Acid catalyzed hydrolysis and Base catalyzed hydrolysis are the contents of the section of Hydrolysis Mechanism and should be merged into Hydrolysis Mechanism.
  4. The authors should compare and summarize which of the two hydrolysis mechanisms is more efficient.
  5. Why can sulfur be further activated by zinc? The authors should have more introductions.
  6. “Zinc ions are known to affect the improvement of the 3D network structure of the silica-silane containing rubber compound, and this is explained by the reaction mechanism of iron ions.” Zinc ions and iron ions have the similar reaction mechanism, and why choose Zinc ions over iron ions?
  7. The performance improvement of Silica-silane filled NR due to the improvement of interfacial adhesion between NR and silicashould be showed by more specific data.
  8. This is true for the using of silane to improve the interfacial adhesionbetween inorganic filler and polymer, see e.g. Biomaterials Research, 2022, 26(1): 1-15, Carbon letters, 2016, 17(1): 79-84. This important information should be integrated into the paper as well.
